# Pharmacokinetic Study and Metabolite Identification of 1-(3′-bromophenyl)-heliamine in Rats

**DOI:** 10.3390/ph15121483

**Published:** 2022-11-29

**Authors:** Ruqi Xi, Rahima Abdulla, Miaomiao Zhang, Zhurakulov Sherzod, Vinogradova Valentina Ivanovna, Maidina Habasi, Yongqiang Liu

**Affiliations:** 1State Key Laboratory Basis of Xinjiang Indigenous Medicinal Plants Resource Utilization, CAS Key Laboratory of Chemistry of Plant Resources in Arid Regions, Xinjiang Technical Institute of Physics and Chemistry, Chinese Academy of Sciences, Urumqi 830011, China; 2University of Chinese Academy of Sciences, No. 19 (A) Yuquan Road, Shijingshan District, Beijing 100049, China; 3S. Yu. Yunusov Institute of the Chemistry of Plant Substances, Academy of Sciences of the Republic of Uzbekistan, Tashkent 100170, Uzbekistan

**Keywords:** pharmacokinetic, metabolites, LC−MS/MS, in silico metabolite prediction, rats

## Abstract

Tetrahydroisoquinolines have been widely investigated for the treatment of arrhythmias. 1−(3′−bromophenyl)−heliamine (BH), an anti−arrhythmias agent, is a synthetic tetrahydroisoquinoline. This study focuses on the pharmacokinetic characterization of BH, as well as the identification of its metabolites, both in vitro and in vivo. A UHPLC−MS/MS method was developed and validated to quantify BH in rat plasma with a linear range of 1–1000 ng/mL. The validated method was applied to a pharmacokinetic study in rats. The maximum concentration C_max_ (568.65 ± 122.14 ng/mL) reached 1.00 ± 0.45 h after oral administration. The main metabolic pathways appeared to be phase-I of demethylation, dehydrogenation, and epoxidation, and phase II of glucuronide and sulfate metabolites. Finally, a total of 18 metabolites were characterized, including 10 phase I metabolites and 8 phase II metabolites. Through the above studies, we have gained a better understanding of the absorption and metabolism of BH in vitro and in vivo, which will provide us with guidance for future in-depth studies on this compound.

## 1. Introduction

Cardiovascular disease, a widespread condition with 80% of the disease burden all over the world, includes arrhythmia, ischemic heart disease, chronic rheumatic heart disease, acute rheumatic fever, pulmonary embolism, heart failure, and hypertension [1,2,3,4]. It is estimated that 17.9 million people die each year from cardiovascular diseases worldwide, and this number is continuously increasing [5,6]. Worldwide, low and middle-income countries, especially China, have the highest burden of cardiovascular diseases [7]. According to the statistics [2], 330 million people suffer from cardiovascular diseases in China, including 4.87 million with arrhythmia. In 2019 alone, the cost of hospitalization for cardiovascular disease in China was CNY 177.338 billion, of which CNY 18.099 billion was spent on arrhythmia [2,8,9]. Currently, antiarrhythmic medicines are clinically classified into four classes, according to their mechanisms [10]: Category I: sodium channel blockers (quinidine, lidocaine, and propafenone) [11]; Category II: β-blockers (propranolol); Category III: anti-fibrillation drugs blocking K^+^ channels (amiodarone); and Category IV: Ca^2+^ channel blockers (verapamil and diltiazem) [12]. 

Tetrahydroisoquinolines are structurally novel calcium channel blockers, similar to dihydropyridines, that act mainly at the dihydropyridine site of the cellular L-type calcium channel [13,14]. Several tetrahydroisoquinoline derivatives, including those from natural and synthetic products, have shown excellent antiarrhythmic effects. For example, tetrandrine [15], berberine [16,17], dauricine [17,18], protopine [19], and other natural products have excellent anti−arrhythmic activity. 1−(3′−bromophenyl)−heliamine (BH) (Figure 1) is a synthetic tetrahydroisoquinoline alkaloid [20]. However, the possible anti-arrhythmic mechanism of BH is as a membrane surface Na^+^–Ca^2+^ exchange inhibitor [21]. In addition, it reduces sarcoplasmic reticulum Ca^2+^ release by inhibiting sarcoplasmic reticulum RyR2Ca^2+^ channels [22]. BH is still undergoing preclinical studies, and its pharmacokinetic profile and metabolites have not yet been studied. Therefore, it is necessary to investigate the pharmacokinetics and metabolites of BH in rats.

The aim of this study is to investigate the absolute bioavailability of BH in rats by both oral and intravenous administration. Subsequently, the metabolites of BH were investigated by in silico, in vitro, and in vivo studies: (a) the in silico prediction of possible metabolic sites of BH by simulating calculation; (b) the in vitro metabolites of BH in rat liver microsomes; (c) the in vivo metabolites of BH in plasma, urine, and feces. The metabolic pathways of BH in rats were proposed, and the pharmacokinetic parameters of BH in rats were summarized.

## 2. Results and Discussion

### 2.1. Pharmacokinetic (PK) Analysis and PK Parameter

The mean plasma concentration profiles of BH after oral and injection administration are shown in Figure 2. The pharmacokinetic parameters obtained by the non-compartmental analysis are summarized in Table 1. The C_max_ of 568.65 ± 122.14 ng/mL was achieved at 1.00 ± 0.45 h after oral administration, indicating that BH was quickly absorbed into the blood circulatory system. The T_1/2_ was 1.62 ± 0.18 h after oral administration, indicating that BH was cleared rapidly from the blood circulatory system. The AUC_0-t_ and AUC_0-inf_ values after oral administration were 1931.81 ± 492.08 and 1968.64 ± 479.81 h*(ng/mL), indicating that the monitoring time was appropriate. The apparent volume of distribution was 2484.43 ± 622.32 mL, indicating a lower plasma protein binding for BH [23]. The absolute bioavailability of BH was 10.6%, indicating low system exposure.

### 2.2. Metabolites of BH

It is crucial for early-stage drug development to conduct metabolic profiling on a new chemical entity. Preliminary in silico metabolism calculations were performed in order to predict possible metabolic sites by XenoSite [24]. Then, in vitro and in vivo metabolites of BH were identified.

#### 2.2.1. Metabolic Sites Prediction of BH

Traditional drug discovery takes 4–5 years, and costs more than USD 200 million for one new drug [25,26]. Even though in silico applications remain limited by shortcomings such as inconsistent and erroneous data, they help us to approach metabolic sites with a higher probability [27]. Figure 3 shows the most likely metabolic sites of BH in human liver microsomes. Figure 3A shows that the tetrahydroisoquinoline part was prone to be hydroxylated. The methoxy part was also highly susceptible to being demethylated. Figure 3B indicates the potential site of epoxidation at the benzene ring. Figure 3C shows the probability of phase II glucuronidation occurring. Figure 3D shows the probability of phase II glucuronidation after demethylation.

#### 2.2.2. Mass Fragmentation of BH

BH is the parent compound of all its metabolites. Since the metabolites derive from BH, the mass spectral characteristics of metabolites show high correlations with those of BH. Therefore, fragmentation of BH has the highest priority in the identification study of metabolites by mass spectrometry. BH possesses a bromine atom, and the natural abundance ratio of ^79^Br and ^81^Br is 1:1. Therefore, either of the two can be selected for further analysis. Herein, ^81^Br was selected for elaboration. The protonated molecular ion peak of BH in ESI positive mode was *m/z* 350.0571 (C_17_H_19_^81^BrNO_2_). The major fragments were *m/z* 333.0304, 252.1145, 179.0943, 170.9628, and 164.0706. The most abundant fragment ion at *m/z* 179.0943 was produced by the loss of a methyl radical and the 3’-bromoband benzene. By further loss of ·CH_3_ (15), the fragment ion *m/z* 164.0706 was yielded. In addition, the C-N bond in the tetrahydroisoquinoline ring was readily cleaved. The fragment ion *m/z* 333.0304 was produced by the loss of NH_3_ (17) from the parent ion. This fragmental ion further generated a daughter ion at *m/z* 252.1145 by loss of a bromine radical. The fragment ion *m/z* 170.9628 was a ring cleavage product. Figure 4 shows the cleavage rule of BH.

#### 2.2.3. Metabolites of BH 

Accurate mass measurements and key fragmental ions of the metabolites of BH in vitro are shown in Table 2. In total, 18 metabolites were characterized, including 10 phase I and 8 phase II metabolites. The surmised pathways of metabolism suggested that BH mainly underwent desaturation, oxidation, dehydrogenation, sulfation, and glucuronidation. The fragmentation processes of M1−A/M1−B, M5, and M6 are shown in Figure 5. Since the isomers of M1−A and M1−B showed the same MS/MS fragment ions one of them was used for illustration in the following structure analysis process. This also applied to M8−A/M8−B, M9−A/M9−B, and M10−A/M10−B. Their accurate mass measurements and key fragmental ions are shown in Table 2. The accurate mass measurements and key fragmental ions of other metabolites not mentioned in the context are also shown in Table 2.

M1−A/M1−B was found to be a demethylated metabolite of BH. It showed a protonated molecular ion peak at *m/z* 336.0416, and its major fragmental ions were observed at *m/z* 319.0154, 286.9890, 238.0988, 170.9628, and 165.0788. The fragment ion *m/z* 170.9628 was a ring cleavage product. The most abundant fragmental ion at *m/z* 165.0788 was generated from the parent ion by the loss of a methyl radical and the 3’-bromoband benzene. In addition, the loss of a NH_3_ from the parent ion led to the fragmental ion *m/z* 319.0154. This fragmental ion further produced daughter ions at *m/z* 238.0988 and 286.9890 by loss of a bromine radical and a methanol moiety, respectively.

M5 was observed with a protonated molecular ion at *m/z* 322.0259. Its major product ion occurred at *m/z* 304.9995, 224.0830, 170.9627, and 151.0628. The fragment of *m/z* 304.9995 was produced from the parent ion by the loss of a NH_3_. It further generated two fragmental ions at *m/z* 224.0830 and 170.9627. The fragment at *m/z* 151.0628 was a ring cleavage product.

M8−A/M8−B was the sulfated metabolite of demethylated BH. It displayed the protonated molecular ion *m/z* 415.9984. The major fragment ions were observed at *m/z* at 336.0416, 319.0148, 286.9890, 238.0987, 170.9627, and 165.0787. The fragmental ion at *m/z* 336.0415 was produced by the loss of sulfate moiety from the parent ion. Afterward, the fragmentations were the same as those of M1−A/M1−B. Similarly, M10−A/M10−B was a glucuronidated metabolite of M5.

M6 gave a precursor ion at *m/z* 364.0370. Its major product ions occurred at *m/z* 190.0863, 175.0628, and 146.0601. By analyzing the base peak m/z 190.0863, the position of epoxidation in the bromobenzene ring was inferred. This assumption was the same as that in in silico prediction (see Figure 3B). The product ion *m/z* 175.0628 was 15 Da less than product ion *m/z* 190.0863, suggesting the subtraction of one methyl from *m/z* 190.0863. The product ion *m/z* 146.0601 was generated from ion *m/z* 175.0628 by the loss of a methoxy as well as electronic rearrangement.

M9−A/M9−B was a multiple-site metabolite of BH. The protonated molecular ion peak *m/z* 508.0437 was observed. Its major fragment ions were *m/z* 332.0103 and 316.9867, and the fragment ion *m/z* 332.0103 was formed by the loss of a glucuronosyl. This fragmental ion further produced daughter ion at *m/z* 316.9867 by loss of a methyl radical. 

The possible metabolic pathways of BH are proposed in Figure 6. These pathways included desaturation, dehydrogenation, oxidation, epoxidation, sulfation, and glucuronidation. Notably, glucuronidated metabolites were found in urine and feces, but not in plasma.

## 3. Materials and Methods

### 3.1. Reagents and Chemicals

BH (97.5%) was obtained from S. Yu. Yunusov Institute of the Chemistry of Plant Substances, Academy of Sciences of the Republic of Uzbekistan (Tashkent, Uzbekistan). Colchicine (93.0%) (Figure 1) was supplied by the National Institute for Food and Shanghai Hong Yong Biotechnology company (Shanghai, China). The UHPLC-MS/MS grades of methanol, acetonitrile, and formic acid were procured from Thermo Fisher Scientific (Bremen, Germany). Blank rat plasma was supplied by the Wuhan purity biotechnology company (Wuhan, China). Ethanol was purchased from the Tianjin Xin platinum chemical company (Tianjin, China). Distilled water was obtained from Watsons (Guangzhou, China). Kolliphor was provided by Sigma-Aldrich (St. Louis, MO, USA). Phase I and UGT metabolic stability kits were purchased from the Beijing Hui Zhi He Yuan biotechnology company (Beijing, China).

### 3.2. Animals and Experiments

All twelve male Sprague–Dawley (SD) rats (weight 200 ± 20 g) were obtained from the Comparative Medicine Center of Xinjiang Medical University (Xinjiang, China). 

BH was formulated in a solution of 2% ethanol, 8% Kolliphor, and 90% distilled water. The twelve rats were fasted for 12 h, but had free access to water, and were randomly divided into two groups. In the oral group, 19.2 mg/kg was orally administered to each rat. Blood was collected from the ophthalmic veins (approximately 0.3 mL) at 0, 0.25, 0.5, 1, 1.5, 2, 4, 8, 12, and 24 h for oral study. In the intravenous group, 1.9 mg/kg was administered by intravenous injection to each rat via the tail. Blood was collected from the ophthalmic veins (approximately 0.3 mL) at 0, 0.1, 0.25, 0.5, 0.75, 1, 1.5, 3.4, 7.5, 12, and 24 h. Then, samples were placed into 1.5 mL polyethylene tubes containing EDTA. Additionally, the blood samples were centrifuged (14,000 rpm) for 10 min. After centrifugation, the supernatant was extracted and then placed at −80 °C for the next step of the analysis.

### 3.3. Calibration Standard and Quality Control Samples in Rat Plasma

Stock standard solutions of BH (1.0 mg/mL) were prepared by dissolving in 0.1 mL of dimethyl sulfoxide and diluting with 30% acetonitrile in water. The first group of stock solution was diluted stepwise into 10, 30, 50, 100, 300, 500, 1000, 3000, 5000, and 10,000 ng/mL of BH calibrator solution. Another set of BH working solutions, with concentrations of 10, 100, 1000, and 5000 ng/mL, were used in the same way for the preparation of quality control samples. The working solution of the internal standard (1.0 mg/mL) was prepared by diluting it with 30% acetonitrile in water. All solutions were stored at 4 °C.

To prepare the calibration standards, the samples were vortexed for 3 s (3000 rpm) with an aliquot of 10 µL BH working solutions in 100 µL blank plasma. Then, 10 µL internal standard was added, and the mixture was vortexed for 3 s (3000 rpm). The mixture was vortexed for 1 min (3000 rpm); afterward, 300 µL acetonitrile was added in order to precipitate the protein. Then, the mixture was centrifuged at 14,000 rpm for 7 min. The supernatant was collected and dried with 40 °C nitrogen. The samples were redissolved with 100 µL of 28% acetonitrile in water (*v*/*v*) and sonicated for 1 min, centrifuged for 7 min, then subjected to UHPLC-MS/MS analysis. Thus, the range of the calibration curve was 1–1000 ng/mL, and the internal standard concentration was 300 ng/mL. In the same manner as described above, quality controls were prepared at four concentration levels. The concentrations of BH and internal standard were 1, 10, 100, 500, and 300 ng/mL. 

### 3.4. Plasma Sample Preparation

A 10 µL volume of 30% acetonitrile in water (*v*/*v*) was added to 100 µL of plasma samples and vortexed for 3 s (3000 rpm). Then, 10 µL internal standard was added to mixtures and vortexed for 3 s (3000 rpm). Subsequently, samples were vortexed for 1 min, after which 300 µL acetonitrile was added for protein precipitation. Additionally, the mixtures were centrifuged for 7 min. The supernatant was collected and dried with 40 °C nitrogen. The samples were re-dissolved with 100 µL of 30% acetonitrile in water (*v*/*v*) and sonicated for 1 min, centrifuged for 7 min, and then subjected to UHPLCMS/MS analysis.

### 3.5. Instruments and UHPLC-MS/MS Conditions

The UHPLC-MS/MS bioanalytical method was established on a Dionex UltiMate 3000 RSL CnanoSystem coupled with Thermo Scientific Q Exactive Plus Orbitrap. The system was controlled by Xcalibur, with data processed using Thermo Xcalibur 4.2.28.14 Quan Browser (Thermo Fisher Scientific, USA).

Chromatographic separation was performed on an Acquity UHPLC BEH C18 (2.1 × 75 mm, 1.7 µm). The column temperature was set at 35 °C. The mobile phase consisted of isocratic formic acid (0.1%): acetonitrile (70:28, *v*/*v*) at a flow rate of 0.2 mL/min. The injection volume was 10 µL, and the autosampler temperature was set at 10 °C. The BH and internal standard were detected by the parallel reaction monitor (PRM) mode. The electrospray ionization (ESI) source parameters were adjusted as follows: automatic gain control (AGC) target, 1 × 10^6^; maximum IT, 100 ms; spray voltage, 3.2 kV; capillary temp, 320 °C S–lens RF level, 55; and high collision dissociation cell (HCD) energy, 35 (for BH) and 38 (for the internal standard).

The UHPLC−MS/MS system and column were the same as those used above. The mobile phase consisted of formic acid (0.1%) and acetonitrile at a flow rate of 0.2 mL/min. The column temperature was set at 35 °C and the injection volume was 10 µL. The gradient elutions were: 0–3 min (5%B), 3–33 min (25%B), 33–60 min (60%B). The metabolites were detected by the full MS/dd MS^2^ mode, with a resolution of 17,500. The heated electrospray ionization (HESI) source parameters were as follows: AGC target, 1 × 10^6^; maximum IT, 100 ms; spray voltage, 3.2 kV; capillary temp, 320 °C; S–lens RF level, 55; and HCD Energy, 20, 40, and 60.

### 3.6. Method Validation

#### 3.6.1. Specificity

The selectivity was evaluated by analyzing blank plasma from six individuals, blank plasma spiked with BH and internal standard, and pharmacokinetic study samples. Blank plasma samples did not exhibit interfering peaks at the retention time of BH or the internal standard. The results are included in Appendix A.

#### 3.6.2. Calibration Curves

A plot of the peak area ratio of BH (y) to the internal standard of the nominal concentration (x) was used to evaluate linearity. The linear range was 1–1000 ng/mL. Additionally, all calibration curves were linear using the least squares regression of the 1/x^2^ weighting factor. The coefficient of determination (R^2^) value was 0.998 for all runs, indicating an adequate linear fit. Data were processed using Thermo Xcalibur 4.2.28.14 Quan Browser.

#### 3.6.3. Accuracy and Precision

Three consecutive batches were determined both on the same day and on three different days to validate intra- and inter-day precision and accuracy. To evaluate the accuracy and precision, four QC concentrations (1, 10 100, and 500 ng/mL) were processed on the same day (intra-day) and on three consecutive days (inter-day). The calculation of relative errors (RE) and relative standard deviations (RSD) was used to determine accuracy and precision. Finally, the intra-day accuracies were in the range of 92.34–105.80%, and the inter-day accuracies were within the range of 96.73–108.20%. The intra-day and inter-day precision values were less than 10%. The results are included in Appendix A.

#### 3.6.4. Recovery and Matrix Effect

The recovery of BH was determined at QC levels in replicates of six. A ratio of BH peak areas from extracted QC samples and blank plasma spiked with authentic standards was calculated. By comparing BH peak areas from spike-after-extraction samples to those from standard solutions at QC levels, the matrix effect was investigated. The recovery and matrix effect of the internal standard was evaluated by the same method at a concentration of 300 ng/mL. The results are included in Appendix A. The recovery and matrix effects of BH from plasma were 93.35–97.13% and 100.01–107.08%, respectively.

#### 3.6.5. Stability

Three QC concentrations (10, 100, and 500 ng/mL) in plasma samples were tested for stability under four conditions: (1) at autosampler for 4, 8, and 12 h; (2) at −80 °C for 1, 7, and 15 days; (3) at room temperature for 12 h; and (4) over three freeze–thaw cycles. In addition, the stability values of BH and internal standard in stock solution were evaluated at 4 °C for 15 days. BH was stable under the above condition. The results are included in Appendix A.

### 3.7. Metabolic Study 

#### 3.7.1. In Silico Metabolism Calculation

The output is displayed as a color scale gradient, distributed on a scale from 0 to 1 (a value of 1 indicates the maximum level of confidence a model can have in predicting the probability that the molecule is a biotransformation site). The metabolic export of CYP is shown in the synopsis form of various human cytochrome isoenzymes (1A2, 2A6, 2B6, 2C8, 2C9, 2C19, 2D6, 2E1, and 3A4). The web-based tool XenoSite was used to assess the metabolism of BH by predicting the atomic sites where BH may be modified by (i) cytochrome P450 [23], (ii) epoxidation [28], and (iii) metabolic modification catalyzed by uridine diphosphate glucuronosyltransferases (UGTs) [29]. XenoSite Web integrates machine learning from neural networks with the computation of several quantitative descriptors of molecules, including their topological and quantum chemical descriptions, as well as the reactivity of atomic positions predicted by Smart-Cyp software [30,31]. The epoxidation precursor unit was developed from the above model, which was applied to the intrinsic reactivity of different molecules with glutathione.

The XenoSite result can be interpreted as a probability that reflects both the model’s confidence about a specific fraction being metabolized and the statistical likelihood that this prediction is correct. In other words, an atom with a prediction score of 0.8 has an 80% chance of becoming an experimentally verified SOM. The output is displayed as a color scale, distributed on a scale from 0 to 1 (the value of 1 reflects the maximum confidence of the model in predicting the probability that the molecule is a biotransformation site). Human cytochrome isozymes (1A2, 2A6, 2B6, 2C8, 2C9, 2C19, 2D6, 2E1, and 3A4) were added to produce the output of CYP metabolism.

#### 3.7.2. In Vitro and In Vivo Experiments 

A rat liver microsome was used the in vitro experiment. For phase I metabolism, the liver microsomes kit contained most of the phase I enzymes. In studies with liver microsomes, the in vitro metabolic system can be reconstituted if the corresponding cofactor, NADPH, is added. Then, phase I metabolic stability studies were performed by the in vitro warm incubation method. The Phase I metabolic reaction operation is as follows: (1) melt all components of the kit in an ice bath and place on ice for use. (2) Mix all components—Liquid A (NADP+, glucose 6-phosphate, and magnesium chloride, (50 μL)), Liquid B (glucose 6-phosphate dehydrogenase and sodium citrate (10 μL)), BH (10 μL), and 0.1M phosphate buffer saline (905 μL))of the incubation system and pre-incubate at 37 °C for 5 min. (3) Add the above mixture (975 μL) to a 2 mL centrifuge tube, preserve the heat in a 37 °C water bath, and then add 25 μL of liver microsome. Blow and suck three times, and mix well to start the metabolic reaction under the conditions of the 37 °C water bath. (4) After 40 min, add 1 mL of precooled acetonitrile to the incubation system to terminate the reaction.

The Phase II metabolic reaction operation is as follows: (1) melt all components of the kit in an ice bath and place on ice for use. (2) Mix all components—Liquid A (NADP+, glucose 6-phosphate, and magnesium chloride, (50 μL)), Liquid B (glucose 6-phosphate dehydrogenase and sodium citrate (10 μL)), UDPGA (50 mM, 100 μL), Alamethicin (250 μg/mL, 100 μL), d-Saccharic acid 1,4-lactone monohydrate (50 mM, 100 μL), BH (10 μL), and 0.1 M phosphate buffer saline (605 μL))—of the incubation system. (3) Add the above mixture (975 μL) into a 2 mL centrifuge tube, preserve the heat in a 37 °C water bath, and then add 25 μL of liver microsome. Blow and suck three times, and mix well to start the metabolic reaction under the conditions of the 37 °C water bath. (4) After 40 min, add 1 mL of precooled acetonitrile to the incubation system to terminate the reaction.

For the plasma study, plasma samples were prepared in the same manner as in the pharmacokinetic study, except that the re-dissolved solvent was methanol. For the urine study, the urine was collected at 0–24 and 24–48 h after administration. Urine from each period was mixed into 10 mL units, and 30 mL of acetonitrile was added. The urine samples were then sonicated for 10 min and aliquoted to 1.5 mL centrifuge tubes. The samples were vortexed for 1 min and centrifuged for 10 min (14,000 rpm), and then the supernatants were merged and dried with nitrogen at 40 °C. The samples were re-dissolved with 100 µL methanol, sonicated for 1 min, centrifuged (14,000 rpm) for 7 min, and then subjected to UHPLC-MS/MS analysis.

For the feces study, the feces were collected at 0–24 and 24–48 h after administration. The samples were dried and de-combined with 8 mL of sample and 40 mL of acetonitrile in a 50 mL centrifuge tube. Then, the samples were sonicated for 30 min. The supernatant was centrifuged at 14,000 rpm for 10 min and blow dried at 40 °C. The samples were re-dissolved with 100 µL methanol, sonicated for 1 min, centrifuged (14,000 rpm) for 7 min, then subjected to UHPLC-MS/MS analysis.

## 4. Conclusions

In conclusion, a composite strategy was proposed to comparatively study the pharmacokinetics of BH. Through the preclinical study, the PK properties of BH were characterized for the first time, which provided helpful information for future clinical research. The mass fragmentation rules of BH were summarized and successfully used to guide the identification of 18 metabolites, both in vitro and in vivo. The metabolic pathways of BH were proposed, including demethylation, dehydrogenation, epoxidation, sulfation, and glucuronidation. Because the in silico prediction can only supply the possible metabolic sites by human metabolic enzymes, it has some limits, even though the metabolic enzymes of rats and humans show some homology. Notably, a high probability of glucuronidation at the “N” site by human metabolic enzymes was predicted by in silico calculations. However, glucuronide metabolite at the “N” site was not found in either the rat (in vivo) study or the rat liver microsome (in vitro) study. This indicates that special attention must be paid to the metabolic differences between humans and rats in future clinical studies.

## Figures and Tables

**Figure 1 pharmaceuticals-15-01483-f001:**
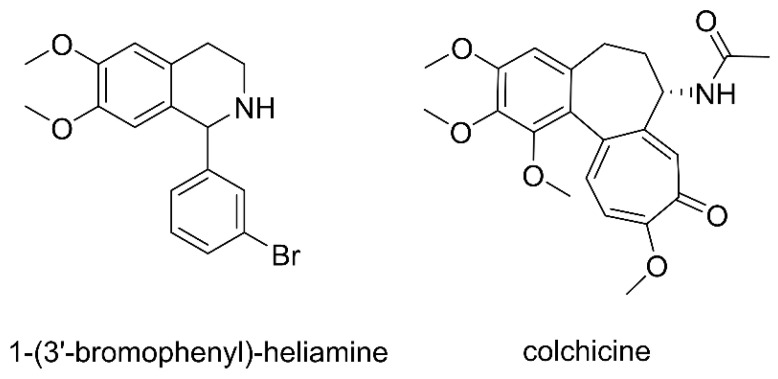
Chemical structures of 1−(3′−bromophenyl)−heliamine and colchicine (internal standard).

**Figure 2 pharmaceuticals-15-01483-f002:**
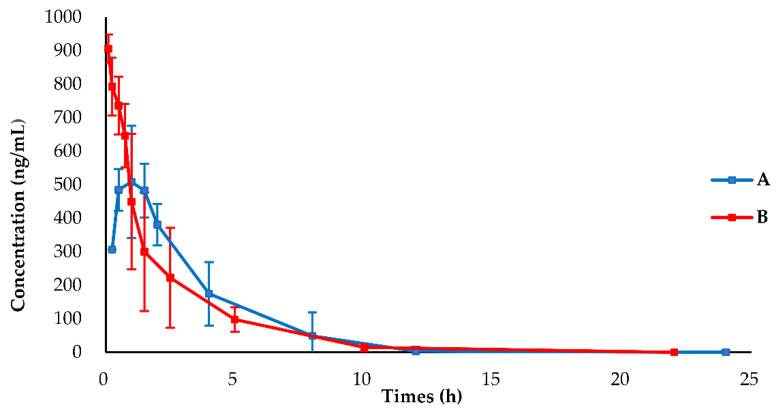
Mean blood concentration vs. time profiles of 1−(3′−bromophenyl)−heliamine after administration to SD rats (*n* = 6), (A) 19.2 mg/kg, orally; (B) 1.9 mg/kg, intravenously.

**Figure 3 pharmaceuticals-15-01483-f003:**
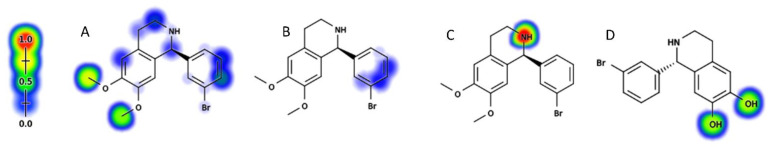
In silico prediction of (**A**) sites of metabolism in human liver microsomes; (**B**) site of epoxidation; (**C**) site of glucuronidation in 1−(3′−bromophenyl)−heliamine; (**D**) site of glucuronidation in O-demethylated metabolite. Potential sites of metabolism are highlighted by a colour gradient, distributed throughout a range of 0 (no colour; minimal probability) to 1 (red colour; maximal probability).

**Figure 4 pharmaceuticals-15-01483-f004:**
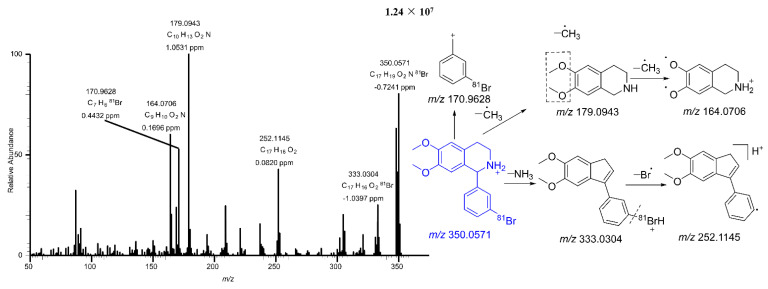
The MS/MS spectra of 1−(3′−bromophenyl)−heliamine.

**Figure 5 pharmaceuticals-15-01483-f005:**
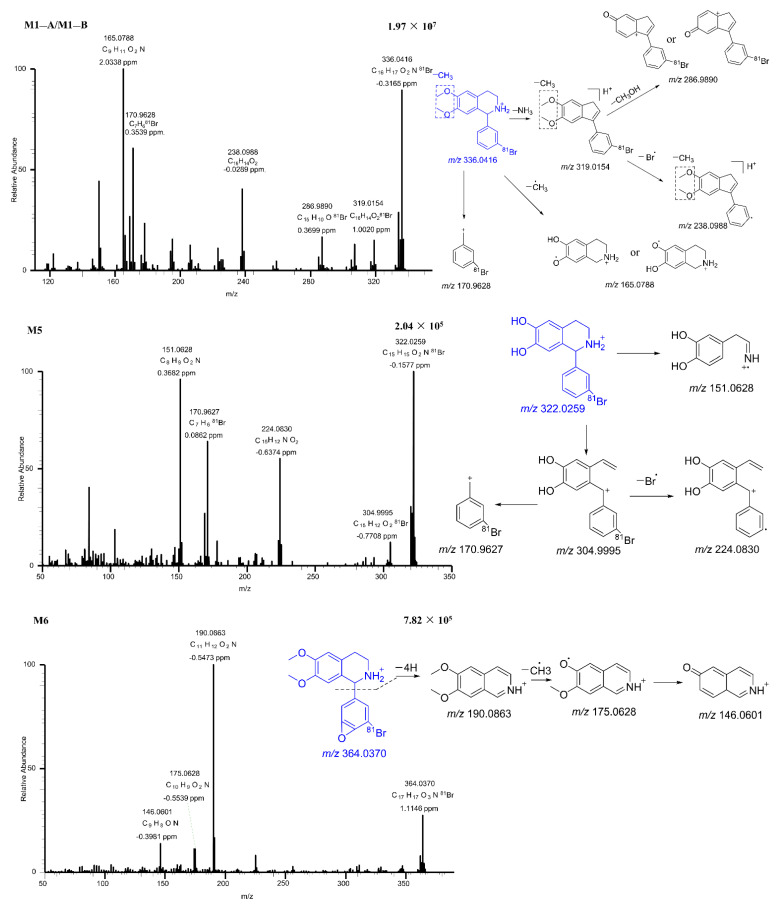
The MS/MS spectra of M1−A/M1−B, M5, and M6.

**Figure 6 pharmaceuticals-15-01483-f006:**
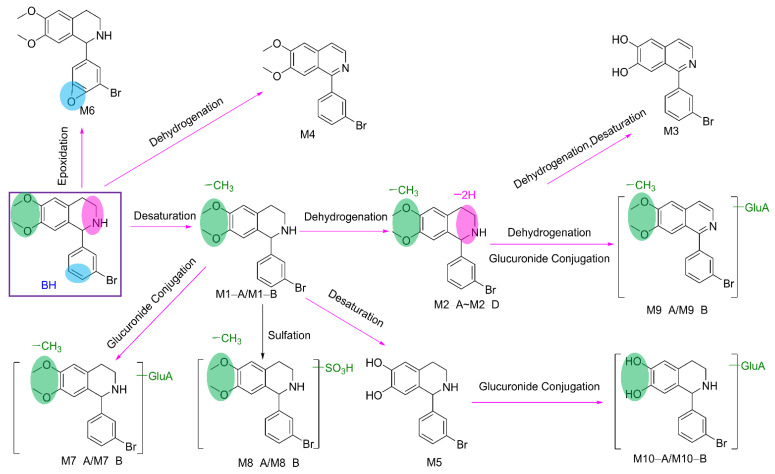
The proposed metabolic pathways of 1−(3′−bromophenyl)−heliamine.

**Table 1 pharmaceuticals-15-01483-t001:** The main pharmacokinetic parameters of 1−(3′−bromophenyl)−heliamine in rats after intravenous and oral administration (*n* = 6, mean ± SD).

Parameters	Unit	PO (19.2 mg/kg)	IV (1.9 mg/kg)
AUC_0–t_ *	h·(ng/mL)	1931.81 ± 492.08	1810.30 ± 696.02
AUC_0–inf_ *	h·(ng/mL)	1968.64 ± 479.81	1902.34 ± 664.42
MRT_0–t_ *	h	2.67 ± 0.38	1.95 ± 0.62
MRT_0–inf_ *	h	2.85 ± 0.24	2.47 ± 0.86
T_max_ *	h	1.00 ± 0.45	—
T_1/2_ *	h	1.62 ± 0.18	2.08 ± 1.01
C_max_ *	ng/mL	568.65 ± 122.14	905.63 ± 46.99
CL *	mL/h	1056.47 ± 202.07	117.25 ± 43
V_d_ *	mL	2484.43 ± 622.32	338.12 ± 167.67
F (%) *	10.6		

* AUC_0–t_ (AUC_0–inf_), area under the analyte concentrations versus time curve from time 0 to t h (inf); MRT_0–t_ (MRT_0–inf_), mean residence time at time 0–t (inf); T_max_, time of maximum concentration; T_1/2_, terminal half-life; C_max_, maximum concentration; CL, clearance; V_d_, apparent volume of distribution; F (%), absolute bioavailability.

**Table 2 pharmaceuticals-15-01483-t002:** Metabolite characterization of 1−(3′−bromophenyl)−heliamine by UHPLC−Q−Orbitrap−MS.

Mouse				Transformations	Fragment Ion	Error (ppm)	Calculated Mass (*m/z*)	Observed Mass (*m/z*)	Formula (M+H^+^)	Retention Time	Metabolites
F *	U *	P *	LM *								
√	√	√	√	Parent	333.0304, 252.1145, 179.0943, 170.9628, 164.0706	0.2	350.0573	350.0571	C_17_H_19_^81^BrNO_2_	28.84	BH
√	√	√	√	Desaturation	319.0154, 286.9890, 238.0988, 170.9628, 165.0788	−0.3	336.0417	336.0416	C_16_H_17_^81^BrNO_2_	18.86	M1−A
√	√	√	√	Desaturation	319.0156, 286.9886, 238.0987, 170.9627, 165.0787	0.1	336.0417	336.0417	C_16_H_17_^81^BrNO_2_	20.88	M1−B
√	√	√	√	Desaturation, Dehydrogenation	319.0022, 176.0707, 165.0789, 150.0551	−0.3	334.026	334.0259	C_16_H_15_^81^BrNO_2_	15.73	M2−A
√	√	√	√	Desaturation, Dehydrogenation	319.0024, 226.0989, 194.0727,176.0707, 161.0473,	−0.7	334.026	334.0258	C_16_H_15_^81^BrNO_2_	17.69	M2−B
√	√	√	√	Desaturation, Dehydrogenation	319.0023, 301.9997, 238.0859	−0.3	334.026	334.0259	C_16_H_15_^81^BrNO_2_	22.82	M2−C
√	√	√	√	Desaturation, Dehydrogenation	319.0024, 301.9998, 238.0858	−0.3	334.026	334.0259	C_16_H_15_^81^BrNO_2_	24.38	M2−D
√	√	√	√	Desaturation, Desaturation, Dehydrogenation, Dehydrogenation	236.0706, 160.0394	−1.0	317.9947	317.9944	C_15_H_11_^81^BrNO_2_	31.6	M3
√	√	√	√	Dehydrogenation, Dehydrogenation	329.9948, 301.9987	2.1	346.026	346.0273	C_17_H_15_^81^BrNO_2_	32.72	M4
√	√	√	√	Desaturation, Desaturation	304.9995, 224.0830, 170.9627, 151.0628	−0.2	322.026	322.0259	C_15_H_15_^81^BrNO_2_	16.52	M5
√	√	√	√	Epoxidation	190.0863, 175.0628, 146.0601	1	364.0366	364.037	C_17_H_17_^81^BrNO_3_	18.64	M6
√	√	×	√	Desaturation, Glucuronide Conjugation	336.0454, 319.0151, 286.9885, 238.0988, 170.9627, 165.0788,	−0.2	512.0738	512.0737	C_22_H_25_^81^BrNO_8_	15.14	M7−A
√	√	×	√	Desaturation, Glucuronide Conjugation	336.0416, 319.0148, 286.9885, 238.0988, 170.9627, 165.0787,	−0.2	512.0738	512.0742	C_22_H_25_^81^BrNO_8_	15.64	M7−B
√	√	×	×	Desaturation, Sulfation	336.0416, 319.0148, 286.9890, 238.0987, 170.9627, 165.0787,	0.4	415.9985	415.9984	C_16_H_17_^81^BrNO_5_S	18.87	M8−A
√	√	×	×	Desaturation, Sulfation	336.0415, 319.0161, 286.9885, 238.0988, 170.9628, 165.0788,	0.1	415.9985	415.9991	C_16_H_17_^81^BrNO_5_S	20.83	M8−B
√	√	×	√	Desaturation, Dehydrogenation, Dehydrogenation, Glucuronide Conjugation	332.0105, 316.9866	−1.8	508.0421	508.0414	C_22_H_21_^81^BrNO_8_	11.7	M9−A
√	√	×	√	Desaturation, Dehydrogenation, Dehydrogenation, Glucuronide Conjugation	332.0103, 316.9867	1	508.0421	508.0437	C_22_H_21_^81^BrNO_8_	18.61	M9−B
×	√	×	√	Desaturation, Desaturation, Glucuronide Conjugation	332.0260, 304.9990, 224.0832, 170.9628, 151.0628	−1.7	498.0581	498.0572	C_21_H_23_^81^BrNO_8_	9.79	M10−A
×	√	×	√	Desaturation, Desaturation, Glucuronide Conjugation	322.0259, 304.9997, 224.0830, 170.9627, 151.0628	0.2	498.0581	498.0582	C_21_H_23_^81^BrNO_8_	12.11	M10−B

* F, feces; U, urine; P, plasma; LM, Liver microsomes.

## Data Availability

Data are contained in the article and Appendix A.

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
