# Peer review of "Pharmacokinetic Study and Metabolite Identification of 1-(3′-bromophenyl)-heliamine in Rats"

_pharmaceuticals, 2022, doi:10.3390/ph15121483_

Round 1

Reviewer 1 Report

The manuscript by Xi et al. reported a study of the pharmacokinetic characterization of anti-arrhythmias agent 1-(3’-bromophenyl)-heliamine (BH) and the identification of its metabolites in rat plasma. This paper also described a UHPLC-MS approach to quantify BH in rat plasma.

The current study is on a topic of relevance to the readers of the journal. I found the writing of this paper can be improved. Please see the comments below.

Major comments:

1.       Please double-check your references. Some titles, authors, or journal abbreviations are not right.

2.       Since your title is more on pharmacokinetic and metabolites identification of BH, you can spend less effort on describing the UHPLC-MS approach and save it for another paper. If you decide to keep this part, do you have a control for BH, such as another tetrahydroisoquinoline? Also, can you give a brief literature summary of current approaches for tetrahydroisoquinoline quantification and their limits?

3.       Line 52, since you mentioned BH is binding to L-type calcium channel on the same site as DHP, why you were mentioning NCX and RyR2 here?

4.       Instead of putting the new UPLC-MS approach (2.1&2.2) in Result and Discussion, you can put it into in Method part. Please describe your approach step by step. How do you prepare samples? What solvent did you use? Which instrument model did you use? Which parameters did you set?

5.       Currently, there are lots of descriptions like “high sensitivity, short peak times”. Can you use data instead? What are your peak times, and how do they compare to other methods? What is your SNR?

6.       You were using all male rats for experiments. Do you think gender will have effects on the BH metabolism?

7.       You have the structures of metabolites in the support information. It might be better to have a figure in the main context showing all the structures of them.

8.       I could not find Figure 6 or any figures in the pdf I received.

9.       Again, if you decide to focus on the BH metabolism, delete the part “A rapid and sensitive UPLC-MS/MS method was developed…”. Alternatively, if you want to focus on the new method in this paper with BH as an example, change your title and Abstract.

Minor comments:

10.   Line17-18. Combine the sentences. Something like “An UHPLC-MS/MS method was developed to quantify the BH level in rat plasma with a linear range from 1-1000 ng/ml.”

11.   Line 29, the abbreviation of “Cardiovascular Disease” is usually CVD. There is nothing wrong with writing the full form instead.

12.   Line29, by definition, “Epidemic” disease is infectious.

13.   Line 32. I checked the paper, 17.9 million was an estimation. You can either say XX million died in 2021 or ref6 estimated XX million people die each year and the number is increasing.

14.   Line 33, move the “low-income and middle-income countries” before “China has …”.

15.   Line 33, please add citations to the statistics.

16.   Line 35, ”Only” is not the best word here.

17.   Line 36, “This imposes…” is not necessary for this introduction.

18.   Please check reference 13. I could not find this title in the database.

19.   Line44, you were mentioning the side effects of available drugs in the previous sentence, I could not see the link to “wider therapeutic window” here.

20.   Line 46-51, Move “Tetrahydroisoquinolines are …anti-arrhythmic activity.”to the beginning of this paragraph.

21.   Line 55-58 “In order to…rats”, can you change it. Instead of focusing on reducing the cost, you can mention that there are no studies on pharmacokinetics of BH in rats currently. Your study will give insights into the mechanisms and reduce costs.

22.   Line 68-71, this part is not related to your results.

23.   Line 74, is UPLC-MS the same as UHPLC-MS you mentioned earlier?

24.   Section6 (Abbreviations)&Section7 (Patents) are not required since you have declared the full forms in the main contexts and there is no real patent. I checked other papers in this journal and did not find other people put Funding under Patents. Please confirm with Style Guide of this journal.

Author Response

We appreciate the reviewers' valuable comments on the writing of this paper. The manuscript has revised following the comments. The English language edited using MDPI editing services (English editing ID: English-54163). All the revisions in the revised manuscript are marked up using the “Track Changes” function in MS word. Herein are detailed responses point by point.

Reviewer 1

The manuscript by Xi et al. reported a study of the pharmacokinetic characterization of anti-arrhythmias agent 1-(3’-bromophenyl)-heliamine (BH) and the identification of its metabolites in rat plasma. This paper also described a UHPLC-MS approach to quantify BH in rat plasma.

The current study is on a topic of relevance to the readers of the journal. I found the writing of this paper can be improved. Please see the comments below.

Major comments:

1. Please double-check your references. Some titles, authors, or journal abbreviations are not right.

Response: Thank you. Writing of all the references is carefully revised one by one following the “Instructions for Authors”.

2. Since your title is more on pharmacokinetic and metabolites identification of BH, you can spend less effort on describing the UHPLC-MS approach and save it for another paper. If you decide to keep this part, do you have a control for BH, such as another tetrahydroisoquinoline? Also, can you give a brief literature summary of current approaches for tetrahydroisoquinoline quantification and their limits?

Response: Thank you very much. Following Comment 4 of Reviewer 1, the description of the quantitative UHPLC-MS approach is shortened and moved to the Method section. Taking into account structural similarity, polarity similarity, and easy availability of standards, colchicine was selected as an internal standard in the mentioned method. There is no literature on the quantification of BH. This manuscript will be the first report on the quantification of BH.

3. Line 52, since you mentioned BH is binding to L-type calcium channel on the same site as DHP, why you were mentioning NCX and RyR2 here?

Response: Sorry, our writing led to this confusion. The introduction has been reorganized and improved.

4. Instead of putting the new UPLC-MS approach (2.1&2.2) in Result and Discussion, you can put it into in Method part. Please describe your approach step by step. How do you prepare samples? What solvent did you use? Which instrument model did you use? Which parameters did you set?

Response: The quantitative UHPLC-MS approach is moved to the Method section and described step by step. For sample preparations see section 3.4. For the solvent system see section 3.5. For the instrument brand and model see section 3.5. For the parameters see section 3.5.

5. Currently, there are lots of descriptions like “high sensitivity, short peak times”. Can you use data instead? What are your peak times, and how do they compare to other methods? What is your SNR?

Response: Thank you. All the descriptions like “high sensitivity, short peak times” have been removed. Alternatively, all descriptions of the quantitative UHPLC-MS method have shown using data. See section 3 and supporting information.

6. You were using all male rats for experiments. Do you think gender will have effects on the BH metabolism?

Response: Due to the estrous cycle of female rats can affect their behavior and lead to tricky results, PK investigations are generally performed with male rats. BH metabolism depends on the metabolic enzymes in the liver, which are gender-independent Then, we think gender will not affect the BH metabolism. Anyway, herein is just a hypothesis with no experimental evidence.

7. You have the structures of metabolites in the support information. It might be better to have a figure in the main context showing all the structures of them.

Response: Thank you very much. All the figures, including the figure for the structures of metabolites, have been shown in the main text. Meanwhile, all the figures upload in a single zip archive following the “Instructions for Authors”.

8. I could not find Figure 6 or any figures in the pdf I received.

Response: Sorry about this. Figure 6 has shown in the revised main text.

9. Again, if you decide to focus on the BH metabolism, delete the part “A rapid and sensitive UPLC-MS/MS method was developed…”. Alternatively, if you want to focus on the new method in this paper with BH as an example, change your title and Abstract.

Response: Thank you. The mentioned part is deleted.

Minor comments:

10. Line17-18. Combine the sentences. Something like “A UHPLC-MS/MS method was developed to quantify the BH level in rat plasma with a linear range from 1-1000 ng/ml.”

Response: Thank you. The mentioned sentences have been combined.

11. Line 29, the abbreviation of “cardiovascular disease” is usually CVD. There is nothing wrong with writing the full form instead.

Response: Thank you. The “cardiovascular disease” is revised to write in full form.

12. Line29, by definition, “Epidemic” disease is infectious.

Response: Thank you. The mentioned part is corrected. See Line 29.

13. Line 32. I checked the paper, and 17.9 million was an estimation. You can either say XX million died in 2021 or ref6 estimated XX million people die each year and the number is increasing.

Response: Thank you. The mentioned part is corrected. See Lines 32-33.

14. Line 33, move the “low-income and middle-income countries” before “China has …”.

Response: Thank you. The mentioned part is revised. See Lines 34-35.

15. Line 33, please add citations to the statistics.

Response: Thank you. The citation is added where appropriate. See Line 37.

16. Line 35,” Only” is not the best word here

Response: Thank you. The mentioned part is revised. See Line 37.

17. Line 36, “This imposes…” is not necessary for this introduction.

Response: Thank you. "This imposes..." has removed.

18: Please check reference 13. I could not find this title in the database.

Response: Thank you. Writing of all the references, including reference 13, is carefully revised one by one following the “Instructions for Authors”.

19: Line44, you were mentioning the side effects of available drugs in the previous sentence, I could not see the link to “wider therapeutic window” here.

Response: Thank you. The introduction has revised. Due to this study not aiming to fix the side effects of available drugs, the mentioned part has rewritten. See line 44-47.

20. Line 46-51, Move “Tetrahydroisoquinolines are …anti-arrhythmic activity. “to the beginning of this paragraph.

Response: Thank you. The mentioned part has moved to the beginning of this paragraph. See line 48-53.

21. Line 55-58 “In order to…rats”, can you change it. Instead of focusing on reducing the cost, you can mention that there are no studies on the pharmacokinetics of BH in rats currently. Your study will give insights into the mechanisms and reduce costs.

Response: Thank you very much. The mentioned part has revised and improved. See Line 58-60.

22. Line 68-71, this part is not related to your results.

Response: Thank you very much. The mentioned part has removed.

23: Line 74, is UPLC-MS the same as UHPLC-MS you mentioned earlier?

Response: All the "UPLC-MS" and the "UHPLC-MS" in the main text are uniformly written as "UHPLC-MS".

24: Section6 (Abbreviations)&Section7 (Patents) are not required since you have declared the full forms in the main contexts and there is no real patent. I checked other papers in this journal and did not find other people put Funding under Patents. Please confirm with the Style Guide of this journal.

Response: Thank you very much. The mentioned parts have removed. The manuscript has thoroughly revised following the “Instructions for Authors”.

Reviewer 2 Report

Work focused on the pharmacokinetic characterization of 1- (3-bromophenyl) heliamine BH which is a synthetic tetrahydroisoquinoline with antiarrhythmic properties. In addition, 18 metabolites of BH were characterized.

Although many interesting data are reported, the work cannot be published in the present form  and must be thoroughly revised.

In particular:

the work is poorly written and organized, English needs to be revised in depth. The figures, although cited in the text, are not present. Also for this reason the work and the results achieved are very difficult to understand. The legends of the tables are missing making the reported data incomprehensible.

The reason why  fragmentation of BH is important for the study and identification of its metabolites  is not  clear at all.

Minor:

Specify BDT in table 1 and 2

Specify DH at page 4 and 5

Table 5 and 6 are missing

Author Response

We appreciate the reviewers' valuable comments on the writing of this paper. The manuscript has revised following the comments. The English language edited using MDPI editing services (English editing ID: English-54163). All the revisions in the revised manuscript are marked up using the “Track Changes” function in MS word. Herein are detailed responses point by point.

Work focused on the pharmacokinetic characterization of 1- (3-bromophenyl) heliamine BH which is a synthetic tetrahydroisoquinoline with antiarrhythmic properties. In addition, 18 metabolites of BH were characterized.

Although many interesting data are reported, the work cannot be published in the present form and must be thoroughly revised.

In particular:

Comment: The work is poorly written and organized, English needs to be revised in depth. The figures, although cited in the text, are not present. Also, for this reason the work and the results achieved are very difficult to understand. The legends of the tables are missing making the reported data incomprehensible.

Response: Thank you very much for the helpful comments on the writing of this paper.

The manuscript has carefully revised.

The English language edited using MDPI editing services (English editing ID: English-54163).

The figures upload in a single zip archive file in the first submission. Herein, all the figures have also shown in the main text in this revised version.

All tables have carefully checked and revised. The missed legends have added.

We think these revisions will make sense for understanding the results achieved.

Comment: The reason why fragmentation of BH is important for the study and identification of its metabolites is not clear at all.

Response: Thank you very much. The identification of its metabolites part has revised. In this revised manuscript, we declare why the Mass Spec fragmentation of BH is important. See Lines 174-177.

Comment:Specify BDT in table 1 and 2; Specify DH on pages 4 and 5;

Response: Thank you very much for your meticulousness. The "BDT" and "DH" were abbreviation mistakes. In this revised manuscript, all the abbreviations for 1-(3’-bromophenyl)-heliamine have corrected to "BH".

Comment: Table 5 and 6 are missing

Response: Thank you very much. All tables have carefully revised and inserted into the main text close to their first citation.

Round 2

Reviewer 1 Report

Thank you for your revision. The manuscript has been improved and I am satisfied.

Author Response

Thank you very much!

Reviewer 2 Report

The manuscript has been extensively revised and reorganized so it can be published as is.

Author Response

Thank you very much!